# Nanoscale Titanium Oxide Memristive Structures for Neuromorphic Applications: Atomic Force Anodization Techniques, Modeling, Chemical Composition, and Resistive Switching Properties

**DOI:** 10.3390/nano15010075

**Published:** 2025-01-06

**Authors:** Vadim I. Avilov, Roman V. Tominov, Zakhar E. Vakulov, Daniel J. Rodriguez, Nikita V. Polupanov, Vladimir A. Smirnov

**Affiliations:** 1Research Laboratory Neuroelectronics and Memristive Nanomaterials (NEUROMENA Lab), Institute of Nanotechnologies, Electronics and Electronic Equipment Engineering, Southern Federal University, Taganrog 347922, Russia; avilovvi@sfedu.ru (V.I.A.); tominov@sfedu.ru (R.V.T.); zvakulov@sfedu.ru (Z.E.V.); srodriges@sfedu.ru (D.J.R.); npolupanov@sfedu.ru (N.V.P.); 2Department of Radioelectronics and Nanoelectronics, Institute of Nanotechnologies, Electronics and Electronic Equipment Engineering, Southern Federal University, Taganrog 347922, Russia

**Keywords:** neuromorphic systems, memristive structure, resistive switching, scanning probe microscopy, local anodic oxidation, titanium oxide nanostructures

## Abstract

This paper presents the results of a study on the formation of nanostructures of electrochemical titanium oxide for neuromorphic applications. Three anodization synthesis techniques were considered to allow the formation of structures with different sizes and productivity: nanodot, lateral, and imprint. The mathematical model allowed us to calculate the processes of oxygen ion transfer to the reaction zone; the growth of the nanostructure due to the oxidation of the titanium film; and the formation of TiO, Ti_2_O_3_, and TiO_2_ oxides in the volume of the growing nanostructure and the redistribution of oxygen vacancies and conduction channel. Modeling of the nanodot structure synthesis process showed that at the initial stages of growth, a conductivity channel was formed, connecting the top and bottom of the nanostructure, which became thinner over time; at approximately 640 ms, this channel broke into upper and lower nuclei, after which the upper part disappeared. Modeling of the lateral nanostructure synthesis process showed that at the initial stages of growth, a conductivity channel was also formed, which quickly disappeared and left a nucleus that moved after the moving AFM tip. The simulation of the imprint nanostructure synthesis process showed the formation of two conductivity channels at a distance corresponding to the dimensions of the template tip. After about 460 ms, both channels broke, leaving behind embryos. The nanodot, lateral, and imprint nanostructure XPS spectra confirmed the theoretical calculations presented earlier: in the near-surface layers, the TiO_2_ oxide was observed, with the subsequent titanium oxide nanostructure surface etching proportion of TiO_2_ decreasing, and proportions of Ti_2_O_3_ and TiO oxides increasing. All nanodot, lateral, and imprint nanostructures showed reproducible resistive switching over 1000 switching cycles and holding their state for 10,000 s at read operation.

## 1. Introduction

Artificial neural networks are one of the main modern trends. The ability to scale them and deep learning allows artificial neural networks to achieve impressive results in various fields [1,2,3,4,5]. Modern neural network algorithms can generate images indistinguishable from photographs [6], conduct text dialogues with a person and answer complex questions [7], recognize graphic and audio information [8], and do much more. In addition, the potential of neural networks is applied in robotics; for example, Tesla cars utilize a fully self-driving computer, and their chip contains a neural network accelerator used to detect a predefined set of objects, including lane lines, pedestrians, and different types of vehicles at a very high frame rate and with a modest budget [9].

The main and significant disadvantage of neural networks is their low performance in modern computers [10,11,12,13]. One of the solutions to this problem is the use of additional accelerators that deal with neural network calculations [14,15,16,17,18]. Such devices allow us to optimize neural network calculations and effectively use computer resources. But the most promising are hardware neural networks in which data processing occurs due to the physical processes of current flow in a passive electrical circuit [19,20,21,22,23].

The hardware implementation of neural networks can be achieved using memristors, while a change in their conductivity during resistive switching can correspond to a change in the “weight” of the neural network [24,25,26,27,28,29]. Among other advantages of the application of memristors, there is a connection of two electrodes used to perform state switching during neural network training and state reading during neural network data processing [30,31,32,33,34,35].

Quantum dots and two-dimensional materials have received significant attention due to their physical, chemical, biological, and memristive properties [33,34]. Along with quantum dots and two-dimensional materials, titanium oxide is well suited for the manufacture of memristors for hardware implementation of neural networks [36,37,38,39,40,41]. Memristor structures based on titanium oxide can switch in a wide range of resistances and allow multilevel switching, which corresponds to a finer adjustment of the neural network “weights”. Often, various “top-down” methods are used to form memristors based on titanium oxide when initially an oxide film is formed and later the lithography is used to produce memristor structures. Such methods include pulsed laser deposition [42], high-frequency deposition [43], deposition of oxide nanoparticles from a solution followed by thermal annealing [44], thermal deposition [45], etc. The disadvantages of these methods include complex lithographic operations for the formation of low-dimensional structures, as well as the need to control the concentration gradient of oxygen vacancies across the thickness of the formed film; for that purpose, additional electroforming or thermal annealing operations can be used.

Other methods of forming memristor structures include bottom-up methods based on selective local growth of oxide nanostructures on the electrode surface due to the influence of catalytic centers or an external electric field [40,41]. In this case, the electrochemical oxidation method allows structures to be created with a nonuniform stoichiometric composition and a reservoir of oxygen vacancies that form a conductivity channel during resistive switching.

This gives rise to the urgent task of conducting further theoretical studies on the mechanism of oxygen vacancy formation in electrochemical titanium oxide nanostructures and conducting experimental studies on resistive switching detection in them.

## 2. Modeling of Oxygen Vacancy Formation Processes in Memristor Nanostructures Based on Electrochemical Titanium Oxide

### 2.1. Analysis of Techniques for Producing Memristor Nanostructures

Despite its advantages, the formation of memristor nanostructures using atomic force microscopy (AFM) methods has several significant drawbacks. Firstly, even though the use of AFM methods allows one to precisely form the dot nanostructures on the substrate surface, their small size can be a problem during subsequent operations of combining the upper contact electrodes. In addition, this technique has low productivity because of the need for sequential formation of dot memristors.

To solve the first problem, the method of lateral displacement of the probe during the formation of the nanostructure can be used. Due to this method, during electrochemical oxidation, the AFM probe scans a given area where the memristor nanostructure will be formed. In this way, we can create a lateral nanostructure of arbitrary size determined by the range of probe movement. However, this method also has low productivity.

To increase the productivity of memristor nanostructure synthesis, we can apply the imprint method, which involves using an imprint template instead of a classical AFM probe and repeating the shape of the required memristor. In addition, it is possible to place several imprint templates in one form, in accordance with the mutual arrangement of individual memristors on the substrate, which will significantly increase the productivity of electrochemical oxidation and allow the entire substrate to be processed in one operation.

### 2.2. Modeling Techniques

Based on the techniques presented, we performed a numerical simulation of the electrochemical titanium oxide dot, lateral, and imprint nanostructure formation processes and calculated the distribution of oxygen vacancies in them (Figure 1). It was assumed that in the case of the dot structure, oxidation occurs under the action of a stationary AFM probe with a curvature radius of 15 nm; in the case of the lateral structure, during the oxidation process, a probe with a radius of 15 nm moved along the substrate surface at a distance of 100 nm; in the case of the imprint structure, a probe with a bottom of 30 nm in length and a curvature radius of 15 nm was applied at the edges. In all cases, the oxidation occurred in 1 s, under a voltage of 10 V applied to the substrate; the feedback system maintained 3 nm from the central point of the probe to the underlying point of the oxide surface; the air humidity in the cell was 90%; and the thickness of the natural oxide layer on the metal surface was 1 nm.

The mathematical model for numerical calculations was taken from the model presented in [46,47], which allows us to calculate the distribution of oxygen vacancies in electrochemical titanium oxide. In this model, a cell containing an AFM probe, a humid air environment, a layer of growing oxide, and a metal substrate was considered. As a result of an applied external voltage influence, oxygen ions are generated in the air environment and transferred through the oxide layer to the metal surface in the region of the chemical oxidation reaction. It was assumed that because of the lack of oxygen at the oxide/titanium interface, the formation of titanium oxide occurs, while additional oxidation processes are also possible in the oxide volume. TiO → Ti_2_O_3_ → TiO_2_. On the basis of the obtained values, the distribution of oxygen vacancies in the formed electrochemical oxide was calculated. Furthermore, further experimental studies showed that the nanostructures obtained by this method exhibit reproducible form-free resistive switching, including multilevel switching [47].

### 2.3. Modeling of the Nanodot Structure Synthesis Process

Numerical modeling showed that in the electrochemical oxidation process, the dot nanostructure was formed, and its height above the surface level increased up to 2.55 nm; the depth of immersion in the metal film was up to 3.11 nm; the total thickness was 5.66 nm; while the diameter of the structure, measured at 10% of the height above the surface level, increased from 56.73 to 67.79 nm (Figure 2a,b).

A detailed analysis demonstrated the distribution of the chemical composition of titanium oxide and oxygen vacancies in the formed nanostructure. The analysis of the results obtained showed that in the initial stage, in accordance with the modeling hypothesis, the natural oxide layer was represented by TiO. Then, as the electrochemical reaction proceeded, TiO was formed at the oxide–metal boundary. As we moved away from this limit, the concentration of TiO decreased since it reacted with oxygen ions from the outside, resulting in the formation of Ti_2_O_3_. In turn, Ti_2_O_3_ was found in the early stages of oxidation both in the volume of the dot nanostructure and near the oxide–air boundary, but then further oxidation by oxygen ions occurred with the formation of TiO_2_. Finally, TiO_2_ was formed near the oxide–air boundary due to the constant supply of oxygen ions to this area and grew deep into the oxide. It is evident from Figure 2 that TiO_2_ did not appear in the early stages of dot nanostructure formation, as it is more probable that TiO will transit to Ti_2_O_3_. Later, approximately 290 ms after the start of the process, TiO_2_ was formed about 15–17 nm from the center of the nanostructure (Figure 2d–g). This can be explained by the fact that the field strength was high in the central part and, consequently, the drift speed of oxygen ions was high and that they did not have time to enter the additional oxidation reaction. As the distance from the center increased, the field strength decreased; thus, oxygen ions drifted more slowly through the oxide, and most of them had time to form TiO_2_ (Figure 2c). As the electrochemical oxidation process continued, the TiO_2_ areas expanded until they closed in the central part approximately 390 ms after the start of oxidation.

Analysis of the oxygen vacancy distribution showed that its location was largely determined by the concentration of TiO and, to a lesser extent, by Ti_2_O_3_. The results demonstrate that throughout the entire electrochemical oxidation process, an oxygen vacancies reservoir was present near the oxide–metal interface, which can subsequently participate in the resistive switching process of the memristor nanostructure. Furthermore, in the initial stages of oxidation, oxygen vacancies formed a conductivity channel connecting the upper and lower oxide boundaries and were determined by the TiO_2_ boundaries. As the TiO_2_ regions were filled, the conductive channel became thinner and the vacancy concentration in it decreased until it broke into upper and lower parts at 640 ms, after which the upper part completely disappeared after another 80 ms and only the nucleus remained at the bottom of the oxide (Figure 2h–k).

### 2.4. Modeling of the Lateral Nanostructure Synthesis Process

As a result of numerical modeling, it was shown that during the electrochemical oxidation process, a lateral nanostructure was formed, and its height above the surface increased up to 1.29 nm, the depth of immersion in the metal film was up to 1.51 nm, and the total thickness was 2.80 nm; conversely, the length of the structure, measured at 10% of the height above the surface, increased from 57.09 to 141.22 nm (Figure 3a,b). It can be noted that, at the initial stages of the local anodic oxidation process, the height of the nanostructure changed significantly, while at later stages, the height remained practically unchanged. In addition, at later stages of formation, the length of the structure increased almost linearly.

This is explained by the technique of nanostructure formation, in which the probe moved linearly along the surface. At the earliest stages of growth, while the probe did not significantly change its position, the formation of the structure proceeded similarly to that of the dot nanostructure; there was a significant increase in height and slow growth in the length of the lateral structure. Then, as the probe moved along the surface, the reaction zone also shifted, with the most intense growth of oxide occurring in front of the probe, where the oxide was thinner and there was a more intense transfer of oxygen ions, while behind the probe, the oxidation slowed down due to a decrease in the flow of oxygen ions due to the large thickness of the oxide (Figure 3c).

A detailed analysis showed the distribution of the chemical composition of titanium oxide and oxygen vacancies in the formed nanostructure (Figure 3d–g). In general, the same mechanism of oxide transition was observed here as was shown earlier. It was assumed that the initial film was TiO, which, under the influence of the flow of oxygen ions, was further oxidized to Ti_2_O_3_; consequently, its concentration in the upper parts of the nanostructure decreased, and its main part was concentrated in the lower part of the nanostructure. Moreover, new TiO was formed during the oxidation process at the oxide–metal boundary, mainly in the region of the oxide growth front (see Figure 3c). The oxide formation was directly associated with a change in the geometry of the nanostructure: a new oxide was formed at its lower boundary and caused an upward shift of the oxide located above that was formed earlier. Thus, the observed movement of the region of increased TiO concentration in the middle part of the nanostructure can be explained as follows. During the probe movement, we observed an intensive oxidation of the metal surface near the probe, with the formation of TiO and its extrusion upward because of the further oxide formation. Then, as the probe passed and moved away, the oxide growth slowed down and the generation of new TiO stopped, but at the same time, oxygen ions continued to drift through this area, which facilitated the transition TiO → Ti_2_O_3_; consequently, the TiO concentration decreased. Similar processes occurred with Ti_2_O_3_, except that this oxide was generated by the additional oxidation of TiO.

The results of the analysis of the oxygen vacancy distribution in the volume of the lateral nanostructure were the same as those of the dot nanostructure. Near the oxide–metal boundary, there was an oxygen vacancy reservoir due to the high concentration of TiO. At the initial stages of nanostructure formation, a conductive channel was formed in its volume, and this channel was later destroyed. The resulting nucleus moved along the oxide following the region with increased TiO (Figure 3h–k).

### 2.5. Modeling of the Imprint Nanostructure Synthesis Process

Numerical modeling showed that in the process of electrochemical oxidation, an imprint nanostructure was formed, and its height above the surface level increased up to 3.60 nm, the depth of immersion in the metal film was up to 4.45 nm, and the total thickness was 8.05 nm; conversely, the length of the structure, measured at 10% of the height above the surface level, increased from 77.86 to 88.73 nm (Figure 4a,b). In this case, the influence of the imprint probe led to the formation of a “plateau”, in contrast to the “peak” for the dot nanostructure. The difference in heights within the imprint probe bottom was in the range of 3.14 to 3.60 nm.

The greater height, compared to the dot nanostructure, can be explained as follows. The increase in the geometric dimensions of the probe led to the fact that high values of the electric field strength were maintained over a larger area under the probe. Consequently, because of the high field strength, a larger number of water molecules would dissociate with the formation of oxygen ions that participate in the oxidation process. In addition, from the presented results (Figure 4c), it is evident that because of the inhomogeneity of the electric field inside the oxide volume, oxygen ions in it will move from the central part to the boundaries of the oxide, which lead to an additional increase in the length of the nanostructure.

A detailed analysis showed the distribution of the chemical composition of titanium oxide and oxygen vacancies in the formed nanostructure (Figure 4d–g). Numerical modeling demonstrated that because of a higher flow of oxygen ions, the TiO concentration dropped sharply, remaining high only near the oxidized surface. A similar situation was observed for Ti_2_O_3_, while the proportion of TiO_2_ increased sharply, filling most of the oxide. In addition, the results show that due to the nonuniform flow of oxygen ions in the oxide volume, TiO → Ti_2_O_3_ → TiO_2_ transitions occurred predominantly in the central part, expanding towards the edge of the oxide and beyond the oxide around the initial oxide film. A sharp change in the oxide composition also indicates an increased generation of oxygen ions in the air. In this case, the nonuniformity of the electric field led to the fact that the oxidation front moved from the central part of the oxide to the edge. Therefore, from the results obtained, it is evident that at the initial stages of growth, two regions were formed along the edges of the nanostructure, in which a high concentration of TiO and Ti_2_O_3_ was maintained for some time; however, as the process continued further, these regions also disappeared (Figure 4d–g).

The analysis of oxygen vacancy distribution showed that during the formation of the imprint structure, two conductivity channels were formed in its volume at once, corresponding to the increase in the concentration of TiO and Ti_2_O_3_, and these channels became thinner over time. At approximately 460 ms, the conductivity channel was broken into two parts, and both parts completely disappeared at approximately 500 ms; as a result, a reservoir of vacancies was formed. It should be noted that the distance between these two channels was approximately 30 nm, which fully corresponded to the bottom of the imprint probe (Figure 4h–k).

The presented results allow us to conclude that any method of forming electrochemical titanium oxide nanostructures results in the formation of a conductivity channel that is later destroyed, leaving a reservoir of oxygen vacancies in the lower part of the oxide. In the case of dot nanostructure formation, a conductivity channel nucleus remains in the central part of the oxide. This nucleus can facilitate the formation of a conductivity channel process in the subsequent stages of resistive switching. Additionally, the same results were obtained for lateral and imprint techniques, with the only difference being that in the lateral nanostructure, the nucleus shifts to the last position of the AFM probe, while for the imprint nanostructure, the nuclei are formed along the entire perimeter.

## 3. Experimental Studies of Resistive Switching in Titanium Oxide Memristive Nanostructures

To confirm the theoretical calculations, experiments on the formation and study of titanium oxide nanostructures using AFM technology were carried out. Using a Solver P47 Pro scanning probe microscope (NT-MDT, Zelenograd, Russia) and HA_FM cantilevers with a conductive W_2_C coating and a tip radius of 15 nm, dot and lateral structure arrays were generated. For this purpose, local anodic oxidation was performed on a thin titanium film (about 30 nm thick) obtained by magnetron sputtering with the application of 10 V voltage pulses between the AFM probe and the titanium film and a feedback circuit current of 0.1 nA. The duration of dot and imprint structure formation was 1 s; the formation of lateral structures was carried out with a 500 × 500 nm raster scan of the AFM probe at a speed of 100 nm/sec. The air humidity in the process chamber was 80%.

As a result, nanodot, lateral, and imprint structure arrays were synthesized on the film surface (Figure 5a–c). To produce the imprint nanostructure, we used a modified AFM probe with a special stamp probe, formed by focused ion beam (Figure 5d). The analysis of the obtained structures showed that when voltage pulses were applied at one point, nanostructures were formed with a height of 2.41 ± 0.10 nm and a diameter of 60.9 ± 9.2 nm. Furthermore, when voltage pulses were applied during raster scanning, lateral nanostructures with a height of 1.16 ± 0.17 nm and lateral dimensions of 576.4 ± 41.2 nm were obtained. And when voltage pulses were applied at imprint probe, nanostructures were formed with a height of 3.35 ± 0.24 nm and a diameter of 2.56 ± 0.03 nm.

### 3.1. Chemical Composition of Titanium Oxide Nanostructures

The titanium oxide nanostructure chemical composition experimental studies were carried out using a ESCALAB 250 spectrometer (Thermo Scientific, Waltham, MA, USA). The energy resolution was determined by the Ag3d5/2 line and was 0.6 eV. As a result, X-ray photoelectron spectra were obtained for the surface of 20 × 20 μm arrays of dots and lateral structures, as well as an imprint structure obtained by combining 25 individual 4 × 4 μm imprint structures. For XPS analysis of the phase composition in depth, ion profiling was used for 30, 60, and 120 s. The analysis of the obtained spectrograms and chemical bonds on the surface and in depth of the thin titanium film and the test titanium ONS was carried out using reference data from the X-ray Photoelectron Spectroscopy Database of the National Institute of Standards and Technology (NIST, Gaithersburg, MD, USA) [48].

The X-ray photoelectron spectroscopy (XPS) measurements detect TiO_2_ at the near-surface layers of all nanostructures: nanodots, lateral, and imprint. The depth profiles following the Ti 2p line consisted of three main components. The first component, with binding energy of about 458.4 eV, belonged to TiO_2_; the second one 456.6 eV—Ti_2_O_3_, and the third one 454.8 eV—TiO (Figure 5e–g). With an increasing depth, the content of Ti_2_O_3_ and TiO at the Ti–substrate surface increased; in this case, the fraction of Ti_2_O_3_ was slightly greater than the fraction of TiO (Figure 5h).

The nanodot, lateral, and imprint nanostructure experimental results confirm the theoretical calculations presented earlier. Under the action of an external electric field, oxygen ions moved from the nanostructure surface to the oxide–metal interface. Due to the low mobility in the oxide, the oxygen ion concentration in the near-surface layers was higher, and the TiO_2_ oxide was observed in the XPS spectra. With the subsequent titanium oxide nanostructure surface etching, according to theoretical calculations, the concentration of oxygen ions significantly decreased. The obtained XPS spectra showed that this region was characterized by a decrease in the proportion of TiO_2_ oxide and an increase in the proportions of Ti_2_O_3_ and TiO oxides. The experimental data correlated well with the theoretical calculations obtained for the point, lateral, and imprint nanostructure (Figure 2d–f, Figure 3d–f and Figure 4d–f).

### 3.2. Resistive Switching of Titanium Oxide Nanostructures

The resistive switching study of the obtained nanostructures was carried out in the spectroscopic measurement mode of the Solver P47 Pro scanning probe microscope. In this case, the conducting AFM probe was positioned at the top of the nanodot or in the middle of the lateral and imprint nanostructure and acted as the upper contact electrode (Figure 6a). To produce an endurance study, the current–voltage characteristics of an individual structure were measured over 1000 switching cycles between the HRS and LRS states in the voltage range of ±4 V (Figure 6b). To study the retention of the LRS and HRS states, the structure first underwent set and reset operations also at voltages of ±4 V to eliminate the influence of the previous state, after which the state was read at a voltage of 0.05 V for 100 s (Figure 6c).

The analysis of the obtained current–voltage characteristics showed that the obtained structures exhibited forming-free resistive switching and over 1000 switching cycles of dot, lateral, and imprint nanostructures; a reproducible effect of resistive switching between the HRS and LRS states was observed (Figure 6d,h,l). Unlike other technological methods, local anodic oxidation allows for the controlled and reproducible creation of oxygen vacancy distributions during the formation of nanoscale titanium oxide structures. Therefore, titanium oxide nanoscale structures demonstrate resistive switching without electroforming [41]. It should be noted that the current–voltage characteristics demonstrated a current limitation within 20 nA, which was conditioned by the built-in current limiter of the SPM.

In all cases, the process of switching to the HRS state (reset) occurs at a positive polarity applied to the metal film and to the LRS state (set) at a negative polarity. This is consistent with the work [49], where this mechanism was explained by the fact that during the resistive switching process under the action of an external electric field, titanium oxide molecules decompose with the release of oxygen ions and the formation of oxygen vacancies. During the drift, oxygen ions can recombine with vacancies at another place in the oxide, resulting in a change in the shape and size of the conductivity channel from oxygen vacancies.

The analysis of the results obtained showed that at a reading voltage of 2.5 V, the resistance of the dot structure in the HRS state was (5.3 ± 2.3) × 10^9^ Ω, and in the LRS state = (1.6.3 ± 0.9) × 10^8^ Ω. The resistance ratio R_HRS_/R_LRS_ was 17.7 ± 11.6 (Figure 6e,f). For the lateral structure, the resistance values in the HRS and LRS states were (9.6 ± 6.8) × 10^9^ Ω and (1.3 ± 0.6) × 10^8^ Ω, respectively. It should be noted that dimensional effects can significantly influence the memristive performance of nanoscale titanium structures. However, detailed analysis of nanoscale size effects is shown in [41]. The resistance ratio R_HRS_/R_LRS_ was 69 ± 48 (Figure 6i,j). For the imprint structure, the resistance values in the HRS and LRS states were (2.9 ± 2.0) × 10^10^ Ω and (4.4 ± 2.8) × 10^8^ Ω, respectively. The resistance ratio R_HRS_/R_LRS_ was 64 ± 26 (Figure 6m,n). It is also worth noting that high endurance and retention was observed at relatively low resistivity ratios. This can be attributed to the fact that at a low HRS/LRS ratio, the number and profile of oxygen vacancies were approximately the same for HRS and LRS, and hence the number of intermediate resistive states was minimized, resulting in higher device stability. Thus, memristors with low HRS/LRS resistance ratio were intentionally fabricated in this work.

The retention study indicated that when the read voltage was applied for 10,000 s, a current of 1.8 ± 0.3 nA in the HRS state and a current of 18.5 ± 0.4 in the LRS state flowed through the dot nanostructure (Figure 6j). At the same time, a current of 0.39 ± 0.11 nA flowed through the lateral structure in the HRS state and a current of 2.05 ± 0.26 nA in the LRS state (Figure 6k). And a current of 0.52 ± 0.02 nA flowed through the imprint structure in the HRS state and a current of 2.16 ± 0.18 nA in the LRS state (Figure 6o). It is worth noting that device-to-device uniformity and stability are important factors of the success of memristive structures manufacturing and application. We believe that cell-to-cell and device-to-device uniformity is determined by the distribution of TiO, Ti_2_O_3_, and TiO_2_ oxides [46].

The results obtained correlate well with the theoretical calculations. During the dot nanostructure formation, a conductivity channel was formed at the initial stages, and it was subsequently destroyed, leaving behind a nucleus. During resistive switching, this nucleus was the base for a new conductivity channel that has greater stability. As a result, higher values of the flowing currents were observed in the HRS and LRS states. At the same time, the lateral and imprint structures were characterized by the presence of a nucleus only at the point of the last probe location (for lateral) or at the nanostructure ends (for imprint), while in the rest of the structure, there was a reservoir of vacancy in the lower part of the oxide. During resistive switching, oxygen vacancy redistribution occurred, which also led to a change in resistance, but in the absence of a nucleus, the current flow was less intense.

## 4. Conclusions

Thus, this paper considered techniques for memristor nanostructure formation based on anodic titanium oxide using a scanning probe microscope. Numerical calculations of the chemical composition distribution of the oxide and oxygen vacancies in dot, lateral, and imprint nanostructures were presented. Numerical modeling of nanodot structure formation showed that, at the initial stages of oxidation, oxygen vacancies formed a conductivity channel connecting the upper and lower oxide boundaries. In the last stages formed, the conductivity channel became thinner until it broke into upper and lower parts at 640 ms, after which the upper part completely disappeared and only the nucleus remained at the bottom of the oxide. Numerical modeling lateral nanostructure formation showed that, at the initial stages of nanostructure formation, a conductive channel was formed in its volume, and this channel was later destroyed. The resulting nucleus moved along the oxide following the region with an increased TiO_x_. The formation of imprint nanostructures by numerical modeling showed that two conductivity channels were formed in their volume at once. The distance between these two channels fully corresponded to the bottom of the imprint probe.

Nanodot, lateral, and imprint nanostructures of electrochemical titanium oxide were fabricated and studied by the XPS method. It was presented that in the near-surface layers, nanodot, lateral, and imprint nanostructures consisted of TiO_2_ oxide. With the subsequent titanium oxide nanostructure surface etching, we observed a decrease in the proportion of TiO_2_ oxide and an increase in the proportions of Ti_2_O_3_ and TiO oxides. The experimental data correlated well with the theoretical calculations obtained for the point, lateral, and imprint nanostructures.

Resistive switching of nanodot, lateral, and imprint nanostructures was performed and studied. The analysis of the obtained current–voltage characteristics showed that over 1000 switching cycles of all nanostructures, a reproducible effect of resistive switching between the HRS and LRS states was observed. The correlation between the results of numerical modeling and experimental studies is shown.

Therefore, anodic oxidation is a method suitable for the fabrication of nanoscale structures based on different materials, and the results obtained can be used in the development of technological foundations for neuroelectronic devices based on electrochemical oxide memristor nanostructures.

## Figures and Tables

**Figure 1 nanomaterials-15-00075-f001:**
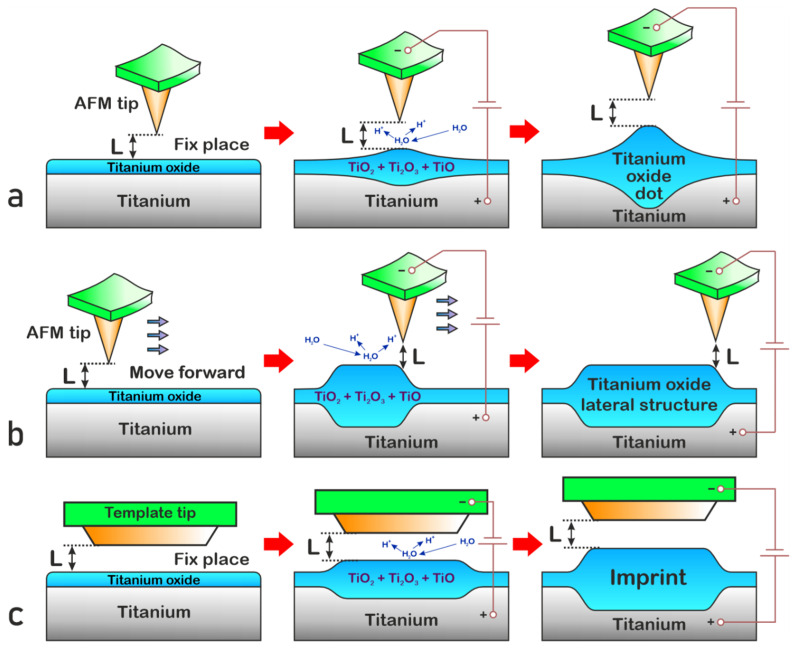
Synthesis techniques of oxide nanostructures by anodization of the titanium film: (**a**) local nanodot, (**b**) lateral, and (**c**) imprint.

**Figure 2 nanomaterials-15-00075-f002:**
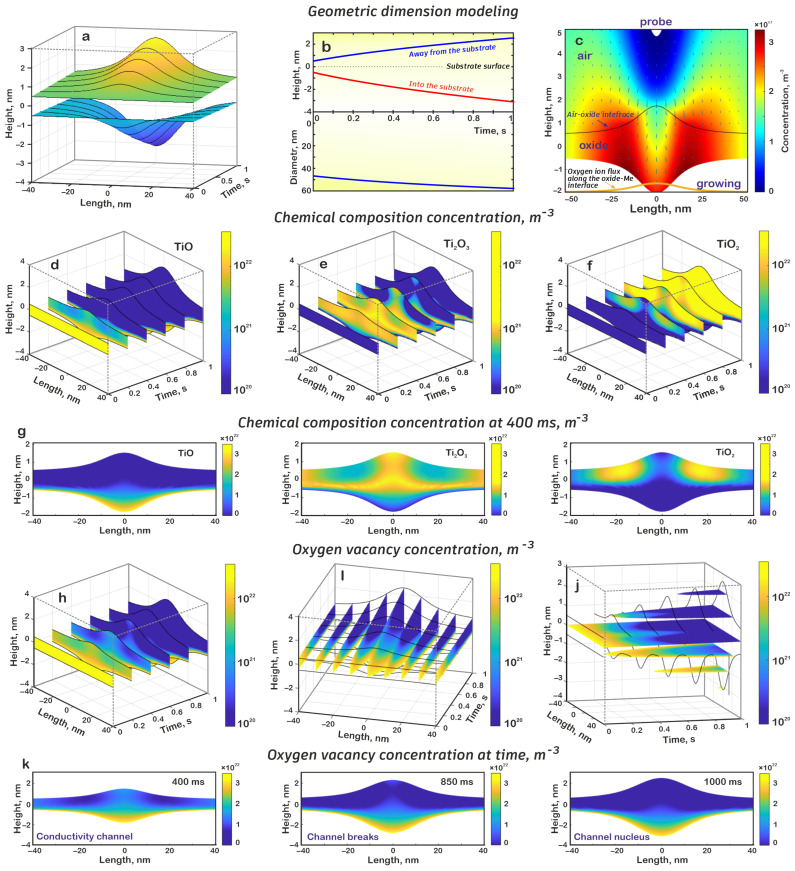
Results of the oxide nanodot structure mathematical simulation: (**a**) dependence of the nanostructure profile on time, (**b**) dependence of the nanostructure height and diameter on time, (**c**) distribution of oxygen ion concentration and electric field vector, (**d**) oxide distribution of the TiO in the nanostructure volume at different moments of time, (**e**) oxide distribution of the Ti_2_O_3_ in the nanostructure volume at different moments of time, (**f**) oxide distribution of the TiO_2_ in the nanostructure volume at different moments of time, (**g**) chemical composition of oxide at 400 ms, (**h**) oxygen vacancy distribution by sections at different moments of time, (**i**) oxygen vacancy distribution by sections at different spatial distances, (**j**) oxygen vacancy distribution by sections at different heights, (**k**) oxygen vacancy distribution concentration at different times.

**Figure 3 nanomaterials-15-00075-f003:**
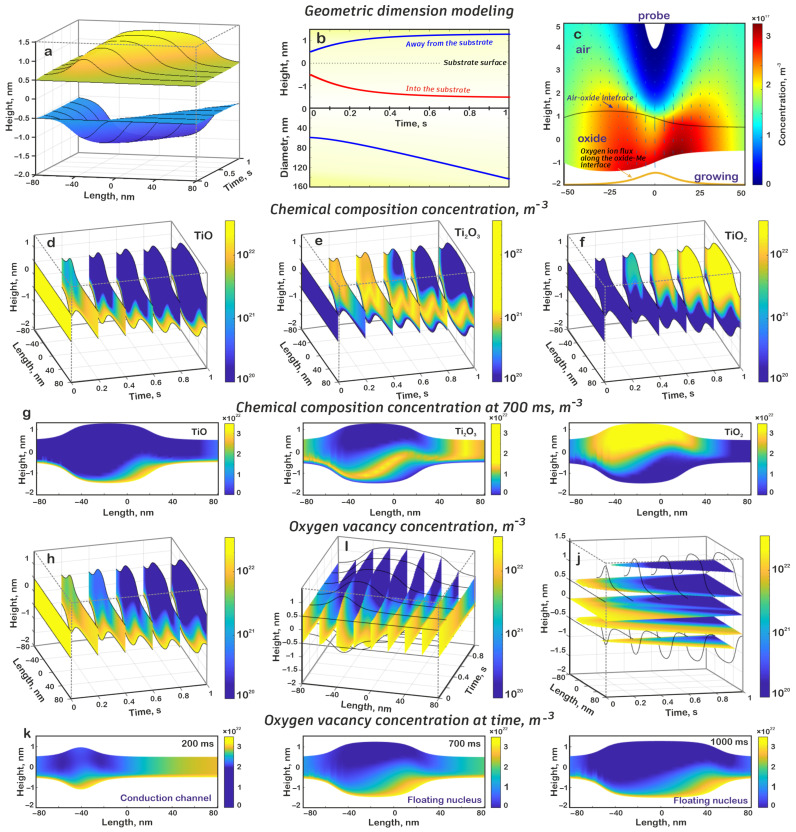
Results of the lateral oxide nanostructure mathematical simulation: (**a**) dependence of the nanostructure profile on time, (**b**) dependence of the nanostructure height and diameter on time, (**c**) distribution of oxygen ion concentration and electric field vector, (**d**) oxide distribution of the TiO in the nanostructure volume at different moments of time, (**e**) oxide distribution of the Ti_2_O_3_ in the nanostructure volume at different moments of time, (**f**) oxide distribution of the TiO_2_ in the nanostructure volume at different moments of time, (**g**) chemical composition of oxide at 700 ms, (**h**) oxygen vacancy distribution by sections at different moments of time, (**i**) oxygen vacancy distribution by sections at different spatial distances, (**j**) oxygen vacancy distribution by sections at different heights, (**k**) oxygen vacancy distribution concentration at different times.

**Figure 4 nanomaterials-15-00075-f004:**
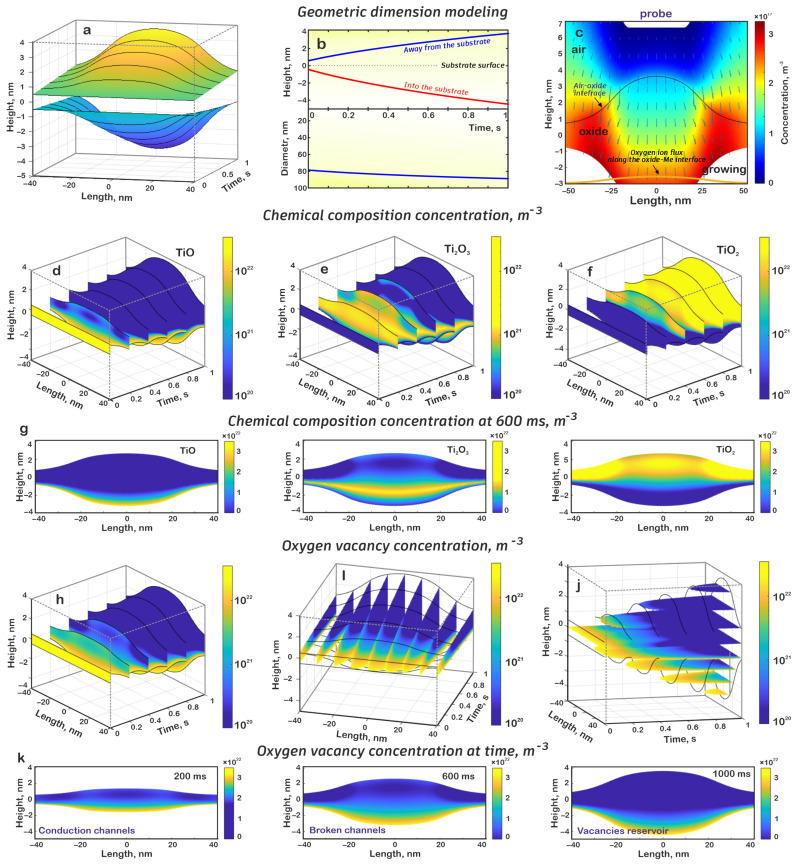
Results of the imprint oxide nanostructure mathematical simulation: (**a**) dependence of the nanostructure profile on time, (**b**) dependence of the nanostructure height and diameter on time, (**c**) distribution of oxygen ion concentration and electric field vector, (**d**) oxide distribution of the TiO in the nanostructure volume at different moments of time, (**e**) oxide distribution of the Ti_2_O_3_ in the nanostructure volume at different moments of time, (**f**) oxide distribution of the TiO_2_ in the nanostructure volume at different moments of time, (**g**) chemical composition of oxide at 600 ms, (**h**) oxygen vacancy distribution by sections at different moments of time, (**i**) oxygen vacancy distribution by sections at different spatial distances, (**j**) oxygen vacancy distribution by sections at different heights, (**k**) oxygen vacancy distribution concentration at different times.

**Figure 5 nanomaterials-15-00075-f005:**
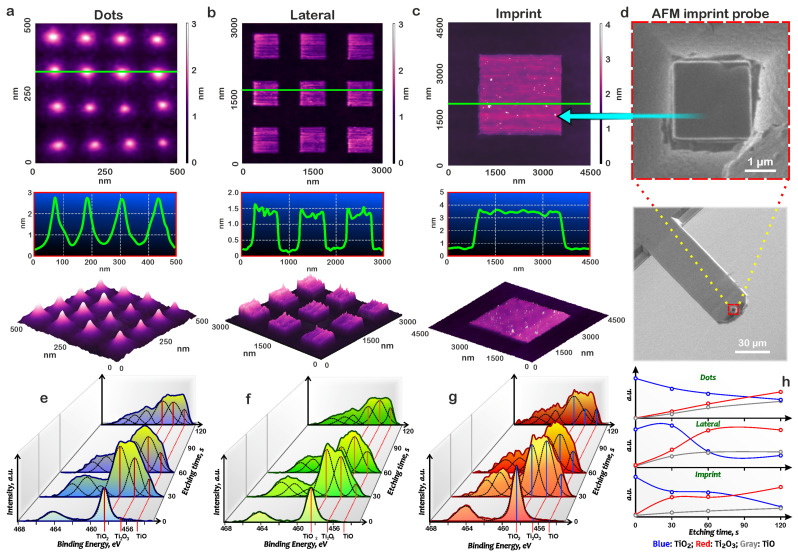
Synthesis and chemical compositions of titanium oxide nanostructures: (**a**) AFM image and profile of nanodot, (**b**) AFM image and profile of lateral nanostructures, (**c**) AFM image and profile of imprint nanostructure arrays, (**d**) SEM image of an imprint probe for the AFM imprint technique; (**e**) XPS spectra of nanodots, (**f**) XPS spectra of lateral nanostructure, (**g**) XPS spectra of imprint nanostructure arrays, (**h**) XPS of titanium oxide distribution by etching time.

**Figure 6 nanomaterials-15-00075-f006:**
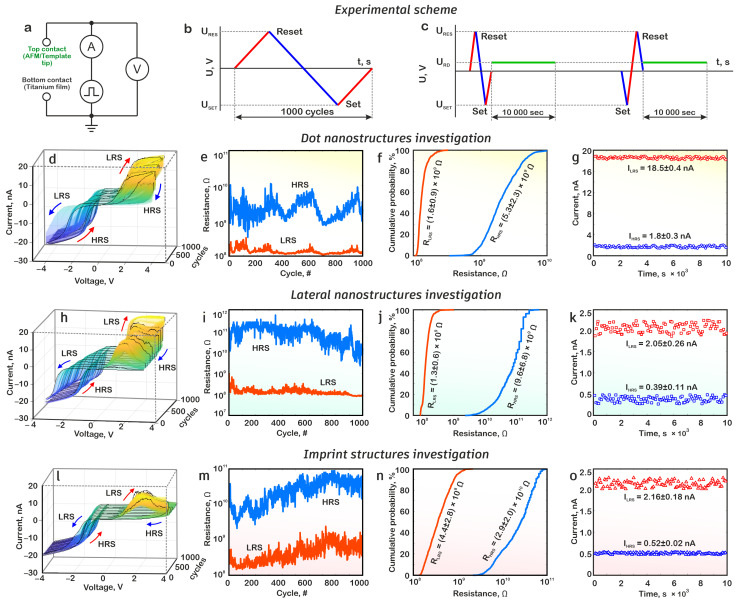
Resistive switching of titanium oxide nanostructures synthesized by different AFM anodization techniques: (**a**) scheme for measuring of electrical characteristics, (**b**) endurance of a nanodot structure and (**c**) retention in the LRS and HRS states, (**d**) resistive switching reproducibility of a nanodot structure, (**e**) endurance of a nanodot structure, (**f**) cumulative probability of a nanodot structure, (**g**) retention of a nanodot structure, (**h**) resistive switching reproducibility of a lateral nanostructure, (**i**) endurance of a lateral nanostructure, (**j**) cumulative probability of a lateral nanostructure, (**k**) retention of a lateral nanostructure, (**l**) resistive switching reproducibility of an imprint nanostructure, (**m**) endurance of an imprint nanostructure, (**n**) cumulative probability of an imprint nanostructure, (**o**) retention of an imprint nanostructure.

## Data Availability

Dataset available on request from the authors.

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
