# Peer review of "Nanoscale Titanium Oxide Memristive Structures for Neuromorphic Applications: Atomic Force Anodization Techniques, Modeling, Chemical Composition, and Resistive Switching Properties"

_nanomaterials, 2025, doi:10.3390/nano15010075_

Round 1
Reviewer 1 Report
Comments and Suggestions for Authors
The manuscript offers a concise and insightful study on the formation of electrochemical titanium oxide nanostructures for neuromorphic applications. By systematically investigating three anodization techniques—nanodots, lateral, and imprint structures—the authors reveal how synthesis methods influence structural and functional properties. The integration of experimental results with a mathematical model adds significant value, providing detailed insights into ion transport, oxide layer formation, and conductivity channel evolution. The alignment between theoretical predictions and experimental observations, including XPS spectra and current-voltage characteristics, strengthens the study’s conclusions. The work is novel in its multi-faceted approach, addressing resistive switching behaviors with impressive reproducibility and stability, which are critical for neuromorphic applications. The following issues need to be addressed before the article can be published:
1. A series of nanomaterials, such as quantum dots and two-dimensional materials, have been employed in the development of memristors. These materials demonstrate promising application prospects in neuromorphic computing. The author should add relevant references (Adv. Mater. 2024, 36, 2405145, Chem. Rev. 2020, 120, 3941)in the introduction section.
2. The authors fabricated three structures of titanium oxide and utilized them in device development. Has the impact of the size of the three structures on memristive performance been studied?
3. Do the three types of devices require a forming process to achieve resistive switching? If so, what is the forming voltage required?
4. Are the data presented in Figure 6 measured from a single device? For memristors, the issue of device-to-device uniformity is critically important. The author should provide data on the uniformity across multiple devices (Matter 2021, 4, 1702).
5. Compared to other reported titanium dioxide memristors, the current ON/OFF ratio of the device is relatively small. Please explain the reason for this. Are there any methods to improve the current ON/OFF ratio?
6. For non-volatile memory devices, a retention time of 100 seconds is too short.
7. Please evaluate whether the fabrication process in this study is applicable to other types of oxides or organic semiconductors.
Reviewer 2 Report
Comments and Suggestions for Authors
This is an excellent study combining careful modelling of the electrochemical formation of TiO2 based neuromorphic lateral structures with experimental results to gain improved understanding on the formation and performance of these structures.
This study can almost be published as is: At the end of page 11 where the ratios of HRS to LRS are reported I was confused about the values for the lateral structure. The are inherently not consistant. Please check these values carefully. I do not see that these ratios should be one order of magnitude higher than for the imprint stuctures. To me they both should be in about the same range.
Round 2
Reviewer 1 Report
Comments and Suggestions for Authors
Agree to accept the manuscript in its current version.